# Inhibitors of Cyclin-Dependent Kinases: Types and Their Mechanism of Action

**DOI:** 10.3390/ijms22062806

**Published:** 2021-03-10

**Authors:** Paweł Łukasik, Irena Baranowska-Bosiacka, Katarzyna Kulczycka, Izabela Gutowska

**Affiliations:** 1Department of Medical Chemistry, Pomeranian Medical University in Szczecin, Powstancow Wlkp. 72 Av., 70-111 Szczecin, Poland; pawel_lukasik@yahoo.co.uk; 2Department of Biochemistry, Pomeranian Medical University in Szczecin, Powstancow Wlkp. 72 Av., 70-111 Szczecin, Poland; ika@pum.edu.pl; 3Department of Pharmaceutical Chemistry, Pomeranian Medical University in Szczecin, Powstancow Wlkp. 72 Av., 70-111 Szczecin, Poland; katarzyna.kulczycka@pum.edu.pl

**Keywords:** cyclin-dependent kinase inhibitors, cancer, cell cycle, CDKs, CDK inhibitors

## Abstract

Recent studies on cyclin-dependent kinase (CDK) inhibitors have revealed that small molecule drugs have become very attractive for the treatment of cancer and neurodegenerative disorders. Most CDK inhibitors have been developed to target the ATP binding pocket. However, CDK kinases possess a very similar catalytic domain and three-dimensional structure. These features make it difficult to achieve required selectivity. Therefore, inhibitors which bind outside the ATP binding site present a great interest in the biomedical field, both from the fundamental point of view and for the wide range of their potential applications. This review tries to explain whether the ATP competitive inhibitors are still an option for future research, and highlights alternative approaches to discover more selective and potent small molecule inhibitors.

## 1. Cyclin-Dependent Kinases (CDKs)

Protein phosphorylation is a necessary mechanism to drive numerous cellular processes such as cell division, migration, differentiation and programmed cell death. This process is regulated by many enzymes, including cyclin-dependent kinases (CDKs) which phosphorylate proteins on their serine and threonine amino acid residues. The 20 members of CDK family known to this day regulate the cell cycle, transcription and splicing [1]. A number of kinase inhibitors are emerging every day as potential small molecule drugs, with some of them already being approved by the United States Food and Drug Administration (FDA). Moreover, these already approved kinase targeting drugs now account for more than a quarter of all available drugs [2]. In relation to CDK inhibitors, drugs such as Palbociclib **1**, Ribociclib **2** and Abemaciclib **3**, have been approved for ER+/HER2- advanced breast cancer treatment [3]. Until recently, the focus of the research was aimed at the highly conserved ATP binding sites of each CDK kinase. Hence, the development of CDK inhibitors has been extremely challenging due to the difficulty of obtaining sufficient selectivity with typical ATP-mimetic compounds. The greatest number of reported compounds has been identified to target the ATP binding pocket. Most recent studies suggest that inhibitors targeting hydrophobic pockets outside the ATP binding site may provide an opportunity for rational target selectivity [4]. Figure 1 illustrates the typical protein structure of the CDK enzyme. The diagram depicts the structural features of a typical kinase domain. Specifically highlighted are the binding pockets of different types of inhibitors, as well as the activation loop.

## 2. Cyclin-Dependent Kinase (CDK) Inhibitors in Drug Development

CDK family is known to regulating the cell cycle, transcription and splicing. Deregulation of any of the stages of the cell cycle or transcription leads to apoptosis but, if uncorrected, it can result in a series of diseases such as cancer or neurodegenerative diseases [1,5,6,7].

Within the last 20 years important advances have been achieved in the development of effective strategies to inhibit CDK kinases. Access of the substrate to the active site of CDK kinase is regulated by the activation loop (A-loop) which is very flexible. The A-loop contains between 20–30 amino acids marked by the conserved Asp-Phe-Gly (DFG) tripeptide motif at the proximal end. Phosphorylation of the activation loop activates the kinase. In this state, the DFG sequence fits snugly into a hydrophobic back pocket adjacent to the ATP binding site. Conversely, in the inactive state the DFG motif swings outwards by partially blocking both the ATP and substrate binding pockets [8]. 

To date, six types of small molecule kinase inhibitors have been defined by the pharmaceutical industry based on their biochemical mechanisms of action (Figure 2). Type I inhibitors interact directly with the ATP binding site and react with the active form of the kinase which is in the DFG-in state and with a phosphorylated activation loop (activation segment). These inhibitors mimic the hydrogen bonds created between the adenine ring of the ATP and the hinge region of the enzyme. Type II inhibitors interact with a DFG-out catalytically inactive conformation of the enzyme and, like type I inhibitors, explore the hinge region and the adenine binding pocket. Type III inhibitors are non-competitive with ATP as they bind to the hydrophobic pocket next to the ATP-binding site, while type IV inhibitors bind away from the ATP binding pocket. Both, type III and IV inhibitors are allosteric in nature [8]. Type V inhibitors interact with two separate regions of the protein kinase domain. This group of inhibitors has been classified as bi-substrate inhibitors. These five classes of inhibitors interact reversibly, while type VI inhibitors form a covalent bond with their target kinase (Figure 2) [9].

## 3. Type I Inhibitors

Many heterocyclic compounds can mimic the hydrogen binding motif of adenine, therefore many type I inhibitors have been discovered. As mentioned above, these compounds interact with the ATP-site of the kinase in its active (DFG-in) conformation and with phosphorylated polypeptide region (activation segment) which lies outside the active site pocket. First generation of structurally diverse ATP competitive small molecule type I CDK inhibitors, produced in the late 1990s and early 2000s, have entered clinical trials to treat numerous solid tumors and hematopoietic malignances. Among the list of compounds that have been synthesized as CDK inhibitors, Flavopiridol (Alvocidib) **4** (Figure 3), a flavonoid derived from an indigenous plant from India, is active against CDK1, CDK2, CDK4, CDK6, and CDK9 with IC_50_ values in the 20–100 nM range (Table 1) [10,11,12,13]. Flavopiridol can inhibit cell cycle progression in G1 as well as G2 phase due to inhibition of CDK2/4 and CDK1 activity, respectively. Early clinical trials proved ineffective because of unsatisfactory efficacy and high toxicity [14,15]. However, later studies confirmed its clinical efficacy in hematological malignancies, and it was granted orphan drug designation for the treatment of patients with acute myeloid leukemia (Table 2) [16].

Roscovitine (Seliciclib) **5** (Figure 3), one of the best known CDK inhibitors, is active against CDK2, CDK5, CDK7 and CDK9 (Table 1). This compound is, by far, the most effective inhibitor of CDK5/p25 (IC_50_ = 160 nM [17]), as shown by numerous studies using this compound as a potential drug against cancer, neurodegenerative or viral diseases, inflammation, polycystic kidney disease (PKD) and glomerulonephritis (Table 2) [18,19,20,21,22]. However, despite many successful preclinical studies, results from several clinical trials are not that promising [23].

Another compound is Dinaciclib **6** (Figure 3), which proved to be a very effective small molecule inhibitor against CDK5 (IC_50_ = 1 nM [24]) (Table 1). Preclinical studies have shown that Dinaciclib is effective against solid tumors and chronic lymphocytic leukemia (CLL), without adversely affecting T-lymphocytes and their numbers (Table 2) [25].

Moreover, Milciclib **7** (Figure 3), an orally bioavailable inhibitor of cyclin-dependent kinases (CDKs) and several other protein kinases responsible for controlling cell growth and replication, has recently obtained the orphan drug designation for thymic carcinoma. It is currently under investigation as a potential drug target for treatment of glioma and hepatocellular carcinoma (HCC) (Table 2) [26,27]. It inhibits CDK2 with IC_50_ of 45 nM and exhibits submicromolar activity against other CDKs including CDK1, CDK4 and CDK5 resulting in a block in the G1 (gap) phase of the cell cycle (Table 1) [28]. Furthermore, Milciclib was found to reduce levels of microRNAs, miR-221 and miR-222, which promote the formation of blood supply (angiogenesis) in cancer tumors [29].

And finally, Palbociclib **1** and Ribociclib **2** (Figure 3), novel CDK4/6 inhibitors, were approved as effective drugs against HR+/HER2- metastatic breast cancer (Table 2) [30,31]. They selectively inhibit CDK4/6 (Table 1), thereby inhibiting retinoblastoma (Rb) protein phosphorylation early in the G1 phase leading to cell cycle arrest, causing defects in DNA replication and efficiently suppress cancer cell proliferation. Most recent data show that both drugs demonstrate a synergistic effect when combined with other drugs, for example Palbociclib and aromatase inhibitor Letrozole [32], Ribociclib and either anaplastic lymphoma kinase (ALK) inhibitor or the mitogen-activated protein kinase kinase (MAP2K, MEK) inhibitor Trametinib [33]. Moreover, utilizing this approach leads to a significant reduction in the development of resistance during prolonged treatment courses [31].

In addition, Tamoxifen **8** has been found to be effective against breast cancer. It reduces CDK5 activity by interacting with p25 and p35, thus preventing CDK5 activation. Tamoxifen can also lower Tau protein phosphorylation, which may suggest that tamoxifen could be used against Alzheimer’s disease [34].

Yet another inhibitor, 5,6-dichlorobenzimidazone-1-β-D-ribofuranoside (DRB) **9** (Figure 3) possesses high selectivity against CDK9, with nearly 25-fold difference in potency over CDK2 and CDK7 (Table 1) [35]. In HeLa cells, DRB (75 μM) inhibited 60-75% of nuclear heterogeneous RNA (hnRNA) synthesis. DRB inhibited a HeLa protein kinase which phosphorylated an RNA polymerase II-derived peptide [36]. DRB can also inhibit HIV transcription (IC_50_ = ~4 μM) by targeting elongation enhanced by the HIV-encoded transactivator Tat (Table 2) [37].

**Table 1 ijms-22-02806-t001:** Kinase inhibitory activities of type I CDK inhibitors.

Inhibitor	Kinase IC_50_ [nM]								
	CDK1/B	CDK2/A	CDK2/E	CDK4/D	CDK5/p25	CDK6/D	CDK7/H	CDK8/C	CDK9/T1
Flavopiridol **4** [38,39]	30	100	100	20–40	-	60	110	-	20
Roscovitine **5** [40]	650	700	700	>100,000	160	>100,000	460	>100,000	600
RO-3306 **17** [41]	35	-	340	>2000	-	-	-	-	-
Dinaciclib **6** [42]	3	1	1	100	1	-	-	-	4
Milciclib **7** [28]	398	45	363	160	265	-	150	-	-
Palbociclib **1** [43]	>10,000	>10,000	>10,000	11	>10,000	15	-	-	-
Ribociclib **2** [44]	113,000	76,000	76,000	10	43,900	39	-	-	-
Abemaciclib **3** [45]	1627	-	504	2	355	10	3910	-	57
BS-181 **18** [46]	8100	730	880	33,000	3000	47,000	21	-	4200
DRB **9** [47]	17,000	-	>10,000	>10,000	-	-	>10,000	>10,000	340
Meriolin 3 **12** [48]	170	11	-	>100,000	170	>100,000	>100,000	-	6
Variolin B **10** [49]	60	80	-	>10,000	90	>10,000	>1000	-	26
Meridianin E **11** [50]	180	800	1800	3000	150	-	-	-	18
Nortopsentins **13** [51]	310–900	-	-	-	-	-	-	-	-
AZD5438 **15** [52]	16	45	6	449	14	21	821	-	20
Roniciclib **19** [53]	7	-	9	11	-	-	25	-	5
SNS-032 **16** [54]	480	38	48	925	340 (CDK5/p35)	-	62	-	4

**Table 2 ijms-22-02806-t002:** Type I CDK inhibitors at different phases of clinical and pre-clinical studies. Trial information obtained from ClinicalTrials.gov as of January 2021.

Inhibitor	Main Targets	Condition or Disease	Phase	Status	Identifier
Flavopiridol **4**	CDK1, CDK2, CDK4, CDK6, CDK9	Acute Myeloid Leukemia (AML)	on the market	“orphan drug”	-
Roscovitine **5**	CDK2, CDK7, CDK9	Pituitary Cushing Disease	II	active	NCT02160730 NCT03774446
		Cystic Fibrosis	II	terminated	NCT02649751
		Advanced Solid Tumors	I	terminated	NCT00999401
		Lung Cancer	II	terminated	NCT00372073
RO-3306 **17** [41]	CDK1	Acute Myeloid Leukemia (AML)	pre-clinical	-	-
Dinaciclib **6**	CDK1, CDK2, CDK5, CDK9	Chronic Lymphocytic Leukemia (CLL)	on the market	“orphan drug”	-
		Breast and Lung Cancers	II	terminated	NCT00732810
Milciclib **7**	CDK1, CDK2, CDK4, CDK7	Hepatocellular Carcinoma (HCC)	II	active	NCT03109886
		Thymic Carcinoma	II	terminated	NCT01301391 NCT01011439
Palbociclib **1**	CDK4, CDK6	HR+/HER2- Breast Cancer	on the market	used in combination with Letrozole	-
			III	active, to be used with other drugs like Fulvestrant	NCT02692755
		Head and Neck, Brain, Colon, and other Solid Cancers	II	active, to be used alone and in combination with different drugs	NCT02255461 NCT03446157 NCT02896335 NCT03965845
Ribociclib **2**	CDK4, CDK6	HR+/HER2- Breast Cancer	on the market	used in combination with Letrozole	-
			III	active, to be used with other drugs like Fulvestrant	NCT02422615 NCT03439046 NCT03294694
		Prostate, and other Solid Cancers	II	active, to be used alone and in combination with different drugs	NCT02555189 NCT01543698 NCT02934568
Abemaciclib **3**	CDK4, CDK6	HR+/HER2- Breast Cancer	on the market	used in combination with Fulvestrant	-
			III	active, to be used with other drugs like Letrozole	NCT02763566
		Lung, Brain, Colon, and other Solid Cancers	II or III	active, to be used alone and in combination with different drugs	NCT04545710 NCT02152631 NCT03220646 NCT04616183 NCT03310879
BS-181 **18** [46]	CDK7	Breast, Lung, Prostate and Colorectal Cancers	pre-clinical	-	-
DRB **9** [55]	CDK7, CDK8, CDK9	HIV Transcription	pre-clinical	-	-
Meriolin 3 **12** [48]	CDK1, CDK2, CDK5, CDK9	Neuroblastoma, Glioma, Myeloma, Colon Cancer	pre-clinical	-	-
Variolin B **10** [56]	CDK1, CDK2, CDK5, CDK9	Murine Leukemia	pre-clinical	-	-
Meridianin E **11** [57]	CDK1, CDK5, CDK9	Larynx Carcinoma, Myeloid Leukemia	pre-clinical	-	-
Nortopsentins **13** [58]	CDK1	Malignant Pleural Mesothelioma (MPM)	pre-clinical	-	-
AZD5438 **15**	CDK1, CDK2, CDK5, CDK6, CDK9	Advanced Solid Malignancies	I	terminated	NCT00088790
Roniciclib **19**	CDK1, CDK2, CDK4, CDK7, CDK9	Lung and Advanced Solid Cancers	II	terminated	NCT02161419 NCT01573338 NCT02656849
SNS-032 **16**	CDK2, CDK7, CDK9	Chronic Lymphocytic Leukemia and other Solid Cancers	I	terminated	NCT00446342 NCT00292864

Novel alkaloids, acting as CDK inhibitors, were also found in some marine organisms. Variolins, 7-azaindole based alkaloids isolated from the antarctic sponge Kirkpatrickia variolosa [59,60], showed in vitro activity against a murine (P388) leukemia cell line with submicromolar potencies by preventing cell proliferation, and inducing apoptosis (Table 2) [56,59]. Variolin B **10** (Figure 3), in particular, was found to inhibit CDK1 and CDK2 kinases, in the micromolar concentration range (Table 1) [61]. Meridianins A-G, a family of 3-(2-aminopyrimidine)indoles, which originate from the ascidian Aplidium meridianum, were demonstrated to inhibit several protein kinases, especially Meridianin E **11** (Figure 3), which can selectively inhibit CDK1 and CDK5 in the low micromolar range (Table 1) [62]. Based on the latter two compounds, Meriolins **12**, a new class of inhibitors, have been designed. These new derivatives have been reported to strongly inhibit various protein kinases, especially CDK1, CDK2, CDK4 and CDK9 (Table 1) [48]. Most recent analysis provides a high potential of Meriolins in the treatment of cancer and noncancer pathologies such as polycystic kidney disease, neurodegenerative diseases, stroke, chronic inflammation, and bipolar disorders (Table 2) [48]. Nortopsentins A-C **13** (Figure 3), antifungal 1,4-bisindolylimidazole marine alkaloids, having an imidazole as a spacer between the two indole units, isolated from the Caribbean deep sea Spongosorites ruetzleri, displayed in vitro cytotoxicity against P388 leukemia cells (IC_50_ 4.5–20.7 µM). Analogues in which the imidazole ring of the alkaloid was replaced by other five or six membered heterocycles were able to inhibit the activity of the cyclin-dependent kinase 1 (CDK1) with submicromolar IC_50_ values (in particular 3-[(2-indolyl)-5-phenyl]-pyridines, phenyl-thiazolyl-7-azaindoles, indolyl-thiazolyl-4-azaindole and indolyl-thiazolyl-7-azaindole derivatives) (Table 1) [51]. Preliminary results indicate, that Nortopsentins, and their analogues, were active against malignant pleural mesothelioma (MPM), a very aggressive human malignancy poorly responsive to currently available therapies (Table 2) [58].

Recent development has enabled combinatorial treatment regimens which can demonstrate synergistic anticancer mechanisms. For instance, THZ1 **14** (Figure 7) a covalent CDK7 inhibitor, was found to selectively downregulate CDK7-mediated phosphorylation of RNA polymerase II, indicative of transcriptional inhibition. Further investigations revealed that the survival of triple negative breast cancer (TNBC) cells relied heavily on the B-cell lymphoma 2 (BCL-2)/B-cell lymphoma-extra large (BCL-XL) signaling axes in cells. Thus, combining the CDK7 inhibitor THZ1 with the BCL-2/BCL-XL inhibitors (ABT-263/ABT199) offer a preclinical proof to significantly improve the poor prognosis in TNBC [63].

However, the complexity of CDK biology and the undesired toxicity related to the off-target effects of the existing pan-CDK inhibitors, led to decisions by several pharmaceutical companies to discontinue the development of many potential anti-cancer agents, exampled with AZD5438 **15**, Roniciclib, SNS-032 **16**, RO-3306 **17**, BS-181 **18** and Roniciclib **19** (Figure 3) (Table 1 and Table 2) [64,65,66]. Therefore, new classes of more selective CDK inhibitors, with strong potential to deliver a meaningful therapeutic impact, were needed. 

One of those compounds is CDK5 inhibitory protein (CIP), a small protein which contribute to nerve cells’ degeneration. CIP specifically blocks the hyperactivated state of CDK5 only when it is linked to p25/p29, while allowing normal activation of CDK5 by p35/p39. The selective inhibition of p25/CDK5 hyperactivation in vivo, through overexpression of CIP, reduced neurodegeneration and improved cognitive function of transgenic mice, without affecting normal neurodevelopment [67]. These findings suggest that CIP could possibly be used to selectively inhibit the p25/CDK5 hyperactivation as a potential therapeutic target to treat certain cancers caused by aberrant CDK5 activation.

## 4. Type 1.5 Inhibitors 

Another strategy to generate novel class of inhibitors has been devised by targeting the inactive unphosphorylated monomeric kinase, not the heterodimer complex. In case of CDK2, a series of compounds based on a quinoline scaffold which bind tightly to the ATP binding site and adjacent back pocket behind the gatekeeper have been synthesized. The binding mode with quinoline-based derivative **20** (Figure 4) in CDK2 demonstrates that the DGF motif is in the “in” state, and **20** interacts not only with the ATP binding site but also disrupts the binding of cyclin A by inducing extensive conformational changes in the C helix (PDB entry 4NJ3). This type of binding to the DFG-in inactive conformation is also referred to as type 1.5 inhibition [68].

Despite showing significant binding affinities none of the compounds exhibited high cellular activity. Quinoline-based **20** bound CDK2 with a K_i_ value of 0.14 µM determined using the CDK2 fluorescence polarization (FP) binding assay and K_d_ = 0.3 µM using the temperature-dependent circular dichroism assay (TdCD). Moreover, the IC_50_ is greater than 10 µM in the CDK2/cyclin A enzyme assay. The explanation might stem from the poor permeability resulting from the carboxyl group, as well as the competition of these inhibitors with cyclin A binding to the monomeric CDK2 (Table 3) [68]. These findings clearly demonstrate the potential of these CDK2 inhibitors. Hopefully, by blocking the interaction with cyclin A, these agents will exhibit different cellular effects which can translate into novel therapeutic possibilities.

## 5. Type II Inhibitors

It has been observed that the CDK active site cleft is very spacious and this fact has been widely exploited in drug discovery. It consists of two regions: the front and back clefts, which are separated by the hydrophobic gatekeeper residue (phenylalanine in nearly all CDK members, methionine in CDK10 and CDK11) [69,70]. The residues necessary to adopt DFG-out conformation are the amino acid at the gatekeeper position and the residue immediately prior to the DFG motif (DFG-1) [71].

The conformational plasticity of the DFG-out binding pocket present a huge opportunity to develop many binding site structural variants which hopefully will be trapped and stabilized by inhibitors [72]. This binding pocket has attracted considerable attention, paving the way for the development of type II inhibitors. Type II inhibitors are anticipated not only to address the problem of kinase inhibitor selectivity but also obtain additional therapeutic benefits such as extended drug target residence times, possess better safety profiles and have fewer side effects [73].

Initially, the development of type II inhibitors had been hampered a little because of the notion that only the simplest amino acids, such as threonine or alanine, as a gatekeeper residue allow the back cleft to be accessible, bulky residues (leucine, methionine or phenylalanine) on the other hand stop a potential small molecule inhibitor from entering the back pocket through this internal gate [74]. Recent studies, however, have shown that kinases with bulkier gatekeeper residues are also able to bind type II inhibitors in the DFG-out state [75]. Moreover, cancerous mutations into larger gatekeeper amino acids generally result in kinase activation, thereby stabilizing the active state of the kinase [76,77,78]. Whether kinases with smaller gatekeeper residues still favor the DFG-out motif has yet to be exemplified.

Although, the factors modulating the DFG-out conformation still remain to be elucidated the initial conclusions can be easily drawn. The results reveal that certain protein kinases such as CDK6, receptor-type tyrosine-protein kinase (RTK, FLT3, CD135), coagulation factor II (thrombin) receptor (PAR1), RAC-b serine/threonine-protein kinase (AKT2), mitogen-activated protein kinase 14 (MAPK14, p38a) and bacterial cell membrane non-specific serine/threonine protein kinase (STK1) favor a classical DFG-out conformation even without the presence of type II inhibitor [79,80], whereas the other inactive, unphosphorylated kinases can be shown to assume the DFG-in conformation [81]. Moreover, Molecular Dynamics (MD) simulations carried out for the Abl tyrosine kinase indicate that DFG binding mode selection might be pH-dependent [82]. Additionally, site-directed mutagenesis (SDM) was used to identify that not only the gatekeeper residue but also the residue located at the N-terminal to the DFG motif play a key role in stabilizing the DFG-out inactive state [71]. Moreover, a comparative analysis of a small library of type II inhibitors showed that over 200 kinases can be targeted, which does not make them intrinsically more selective than type I inhibitors [8]. Moreover, a number of kinases, bound to a type II inhibitor, can exhibit many intermediate states of the DFG-in and DFG-out conformations [83]. The advantages of knowing which and how many enzymes may be targeted by type II inhibitors will be of great value.

Alanine is the most frequently observed amino acid residue at the DFG-1 position (Ala144 in case of CDK2) [84]. However, more data is needed to demonstrate its role in the stabilization of the DFG-out state as one group stated that mutating leucine to cysteine at the DFG-1 position in Mitogen-activated protein kinase 1 (MAPK1) makes it impossible to bind a type II inhibitor by disrupting the DFG-out state [71], while the other one showed that by changing alanine to either cysteine or glycine seem to participate in the stabilization of the DFG-out conformation in CDK2 [85]. Moreover, there are protein kinases which have a cysteine at this position, such as Mitogen-activated protein kinase kinase kinase 7 (MAP3K7, TAK1) which hopefully could bind type II inhibitors [86]. Until more recent information becomes available, it is worth noting that each type of protein kinase should be considered individually, and be limited to the specific case of particular type II inhibitor structure [85].

First attempts to synthesize type II inhibitors of CDK2, the most studied CDK kinase, were based upon their endogenous inhibitors such as the **IN**hibitors of CD**K4** (INK4) (p16 and p18) and the CDK interacting proteins/Kinase inhibitory proteins (Cip/Kip) (p21, p27 and p57). The structural studies focused on the interaction of p27 with the CDK2 N-terminal lobe and the cyclin A box revealed that p27 inserts itself into the ATP binding site, thus preventing its conformational activation (PDB entry 1JSU) [87]. The INK4 family inhibitor p18 in the p18–CDK6/cyclin K ternary complex was also found to inactivate the CDK/cyclin dimer structure by distorting the ATP binding site and misaligning catalytic residues (PDB entry 1G3N) [88]. These observations support the model that the other CDKs may undergo similar inhibitory conformational changes by binding to their respective CDK inhibitors. Numerous peptides and peptidomimetics, based on the sequence alignment of the cyclin-binding motif found in many CDK inhibitory proteins (especially p21 and p27), have been synthesized. Two 8-amino-acid oligopeptidic units H-His-Ala-Lys-Arg-Arg-Leu-Ile-Phe-NH_2_
**21** and H-Ala-Ala-Abu-Arg-Arg-Leu-Ile-*p*FPhe-NH_2_
**22** showed the highest growth-inhibitory activities in both the cyclin A competitive binding assay and the CDK2/cyclin A kinase functional assay with IC_50_ values in the low nanomolar region (Table 4) [89,90]. Another example of the peptidomimetic molecule is MM-D37K **23**, derived from p16, which was found to be the first cyclin D-CDK4/6 alternative class inhibitor in the clinic for colorectal cancer. Collected data will certainly allow interesting comparisons with existing type I inhibitors [91].

The first small molecule inhibitor which induced the DFG-out motif was Sorafenib **24** (Figure 5), a well-known multikinase type II inhibitor, bound to CDK8/Cyclin C heterodimer complex (PDB entry 3RGF). In case of CDK8 the DFG motif is replaced by a unique Asp-Met-Gly tripeptide motif (DMG) [93]. Sorafenib has been found to disrupt mitogen-activated protein kinase (Ras-MAPK) signaling in many cell-based assays, such as colon, liver, kidney, lung, and breast cancer cell lines [95,96]. The function of Ras-MAPK pathway is to transduce signals from the extracellular receptor to the DNA in the cell nucleus where specific genes are activated for cell growth, division and differentiation [97]. Structure-guided modification of Sorafenib resulted in a series of potent CDK8 inhibitors stabilizing the DMG-out conformation such as compound **25** (Figure 5). However, these inhibitors (including Sorafenib) demonstrated weak activity in cellular assays (they neither suppressed the Wnt/β-catenin pathway nor phosphorylated Signal transducer and activator of transcription 1 (STAT1) at Ser727). These findings also suggest that type II inhibitors target the inactive form of CDK8 which is poorly accessible in cells due to the fact that it either forms the Mediator or the kinase-module [94,98]. Most recent data show that the antitumor efficacy of Sorafenib can be enhanced by the addition of Flavopiridol in both Sorafenib-sensitive and Sorafenib-resistant hepatocellular carcinoma (HCC) cell lines. The enhancing effects result from the synergistic effect of co-targeting two different biological mechanisms: CDKs (Flavopiridol) and the Ras-MAPK pathway (Sorafenib), both being linked to the suppression of Mcl-1 expression [99].

Another example of inhibitor able to adopt the DFG-out conformation is K03861 (AUZ454) **26** (Figure 5) an aminopyrimidine-phenyl urea inhibitor. This is a type II CDK2 inhibitor with K_d_ values of 50 nM, 18.6 nM, 15.4 nM, and 9.7 nM for CDK2 (wild type), CDK2(C118L), CDK2(A144C), and CDK2(C118L/A144C), respectively (Table 4). The co-crystal structure of K03861 bound to cyclin-free CDK2 exhibit a type II binding mode with the DFG-out state (PDB entry 1b38). Further analysis of this compound, obtained from kinetic binding experiments, revealed slow off-rates, meaning that compounds exhibiting slow dissociation rates could be considered as a clinically important and statistically significant benefit to patients since they interact with a kinase for much longer [85].

## 6. Type III Inhibitors

This type of inhibitors are compounds which make specific interactions with an exclusive pocket, known as the back pocket of the kinase, which is adjacent to the ATP binding site. Type I/II kinase inhibitors are very sensitive to the gatekeeper mutations affecting the residues within the ATP pocket, the region that has been recognized as responsible for acquired resistance to type I and II kinase inhibitors [100,101].

PD184352 (CI-1040), a selective oral mitogen-activated protein kinase kinase 1/2 (MAP2K1/2, MEK1/2) inhibitor, was the first type III inhibitor to enter clinical trials, that laid the groundwork for the discovery of additional non-ATP-competitive inhibitors [102]. However, no such molecule has been reported in case of type III CDK inhibitors.

## 7. Type IV Inhibitors

Type IV inhibitors have been defined as compounds which bind to unique structural features remote from the ATP binding pocket and are able to interact with these allosteric regions by stabilizing inactive conformations. The allosteric pocket of type IV inhibitors can be located anywhere within the kinase, with one exception for the hydrophobic pocket close to the ATP-binding site which is targeted by type III inhibitors [103]. Potential compounds able to allosterically regulate kinase enzymatic activity will offer the possibility of achieving distinctive advantages which could make them very valuable. Type IV inhibitors do not need to interfere with the phosphorylation of all native substrates but only some, allowing them to block the kinase functions responsible for a particular disease but at the same time preserving their positive functions. However, it is still very difficult to determine what sites are necessary for a certain biological function. The arduous investigation to predict potential allosteric kinase hot spots identified ten different sites outside the ATP site that can be utilized in future development of type IV kinase inhibitors, and their applications in regulating kinase activity in a variety of disease states [4,104,105].

Based on structural features of CDK2, a novel allosteric inhibitor, 8-anilino-1-naphthalene sulfonate (ANS) **27**, was discovered (Figure 6).

It was found to bind to monomeric CDK2 by exploring a cavity very close to the DFG region which results in a structural transformation able to disrupt interactions with the CDK2 activator cyclin A. The activation loop adopts the active DFG-in conformation (PDB entry 3PXF). Consistent with its weak binding affinity to CDK2 (K_d_ = 37 µM) ANS can be easily displaced from the enzyme by cyclin A with an EC_50_ value of 0.6 μM. In addition, ANS was found to inhibit the active, phosphorylated CDK2/cyclin A dimer complex with a poor IC_50_ value of 91 μM (Table 5). It has been concluded that inhibitors with an ANS-like binding mode must interact more efficiently with monomeric CDK2 to noticeably improve their binding affinity, in order to inhibit complex formation with cyclins [106].

## 8. Type V Inhibitors

The preparation of type V inhibitors is considered as a new method to discover compounds which target both the ATP-binding site as well as distinct structural elements unique to each protein kinase in order increase their potency and selectivity. This group of compounds refers to bi-substrate inhibitors. However, the key problem relating to this class of agents is maintaining a balance between potency and selectivity in order to modulate their cellular activity or physicochemical properties [107,108]. A series of highly selective and potent type V inhibitors targeting tyrosine and serine/threonine kinases have been synthesized [109,110,111,112], but as of yet, no such molecule has been reported to be active against CDK family members.

## 9. Type VI Inhibitors

In recent years, there has been rapid progress made in the development of kinase inhibitors which can make covalent, very often irreversible, bond with the kinase active pocket (Figure 7).

The initial research findings, relating to the covalent-binding drugs, indicated that some of these agents can be beneficial to our health and some not necessarily. For instance, aspirin binds with a serine residue of the cyclooxygenase-1 (COX-1) enzyme, by forming covalent adducts, hence preventing the production of proinflammatory cytokines [113]. In contrast, paracetamol metabolizes into highly reactive radicals, although only in about 3%, when overdosed can cause oxidative stress, by forming toxic covalent adducts with liver proteins [114]. However, the most recent advances in Computational Biosciences have made it possible to design compounds with augmented selectivity and efficacy, and limited adverse effects.

Type VI inhibitors utilize chemical properties of type I-IV inhibitors, but they possess additional electrophilic groups (known as warheads) which mainly react with a nucleophilic cysteine residue in the active site (occasionally they also target lysine and tyrosine residues). Irreversible kinase inhibitors are meant to limit drug resistance given by protein kinase mutations, as well as overcome the competition from endogenous ATP. The adduct is generated in the Michael reaction through an acrylamide group (electrophilic warhead) which favors the formation of bonds with cysteine residues. It is thought that by lowering the reactivity of the warhead of type VI inhibitor to generate new shorter-acting reversible type IV agents could reduce their toxicity and off-target reactivity [115].

The first irreversible kinase inhibitor is THZ1 **14** (Figure 7) which covalently binds to a cysteine residue (Cys312) in the ATP binding pocket (PDB entry 1UA2). This compound inhibits CDK7. At higher concentration it also demonstrates some activity against closely related kinases CDK12 and CDK13 (Table 6) [116]. Based on the THZ1 scaffold a new more selective covalent inhibitors, SY-1365 **28** and THZ531 **29** (Figure 7), have been identified. SY-1365 is currently being investigated for the treatment of ovarian and breast cancers (NCT03134638) [117], whereas THZ531 turned out to be a selective covalent inhibitor of CDK12/13 [118] (Table 6). To further optimize the structure of THZ1-like inhibitors another generation of irreversible CDK inhibitors, E9 **30** (Figure 7), has been proposed. The findings show that E9 can overcome a common problem of resistance to the THZ1-like agents by ABC transporter-mediated drug efflux, and it covalently targets CDK12 (Table 6) [119].

Comparative studies of type VI inhibitors targeting other kinases, such as Ibrutinib or Afatinib, with type I and type II inhibitors demonstrated long-term clinical benefits of early treatment of patients with chronic lymphocytic leukemia (CLL), small lymphocytic lymphoma (SLL) [120] and lung cancer [121]. However, the research aimed at targeting only one amino acid (cysteine) can lead to a single point mutation resulting in acquired resistance to this particular agent. Therefore, the latest studies are focused on the development of type VI CDK inhibitors able to utilize the reactivity of other nucleophilic amino acids, such as lysine, tyrosine or even aspartic acid residues [115,122]. Hopefully, this theoretical data will soon generate new type VI CDK inhibitors.

## 10. Targeted Protein Degradation (TPD)

Recent advances in medical modalities gave rise to an appealing and promising technology known as Targeted Protein Degradation (TPD). TPD is a highly efficient method for selectively targeting proteins for removal from the cell, rather than inhibiting their activity. It is anticipated that, by using this method, toxic and disease-causing proteins could be depleted from cells under the potentially effective low-dose treatment. Small molecules able to induce degradation of target proteins can be divided into three major classes based on the structure of the compounds and their mechanism of action [123]. Single-ligand molecules able to create a direct interaction with the target protein to induce degradation belong to the first class of compounds. This group of compounds is represented by the aforementioned Fulvestrant, a selective estrogen receptor downregulator (SERD) which reduces the estrogen receptor-α (ERα) protein level [124]. However, this approach is limited to the finite number of target proteins. 

Compounds that interact with E3 ubiquitin ligase to modulate substrate selectivity to modulate substrate selectivity are known as E3 modulators or molecular glues. The processes by which degradation of proteins is induced include: ubiquitination, targeting to the proteasome, proteolysis and functional silencing. Molecular glues act sub-stoichiometrically to facilitate rapid depletion of previously inaccessible proteins, but have mostly been identified somewhat serendipitously [125]. The first molecular glue was thalidomide which was identified to interact with *CRBN* gene (Cereblon) [126], a substrate recognition subunit of the Cullin-RING E3 ubiquitin ligase (CRL4) [127].

In relation to CDK molecular glues, the first compound acting as molecular glue degrader is *R*‑CR8 **31**, a pan-selective cyclin-dependent kinase (CDK) inhibitor (very similar to *R*-Roscovitine) (Figure 8) [128]. *R*-CR8 binds to the CDK12/cyclin K dimer, the resultant surface-exposed 2-pyridyl moiety facilitates CDK12/cyclin K complex formation with DDB1, the CUL4 adaptor protein, by circumventing the necessity for a substrate receptor and triggers rapid proteasomal degradation of cyclin K [129].

The third class encompasses the chimeric small molecules, where an E3 ligase component and a protein of interest are linked to form a new and unique molecule. This group of compounds was developed under different names such as PROteolysis TArgeting Chimeras (PROTACs) and Specific and Non-genetic IAP-dependent Protein Erasers (SNIPERs). They target different proteins, but their mechanism of action is almost identical. Both, PROTACs and SNIPERs initiate the degradation of targeted protein by linking the protein of interest to an E3 ubiquitin ligase using the cell’s natural ubiquitin proteasome pathway (UPS) [130]. 

In relation to CDK kinases a series of PROTECs molecules have been reported. First dual CDK4/6 degraders **32**, synthesized by linking Pomalidomide and Palbociclib, were reported by Burgess which efficiently degraded CDK4/6 with DC_50_ values ranging from 20–50 nM (Figure 9). 

However, these compounds were not active in cells with overexpressed CDK4/6 [131]. Another group identified both dual CDK4/6 degraders **33** (based on Thalidomide and Palbociclib), as well as selective CDK4 **34** (based on Thalidomide and Ribociclib) and CDK6 **35** (based on Thalidomide and Palbociclib) degraders (Figure 9). These compounds exhibited good target degradation at 100 nM and showed more profound antiproliferative activities [132,133]. Very promising CDK6 degrader **36** was synthesized by linking Pomalidomide and Palbociclib (Figure 9). It possessed high CDK6 degradation capacity with a DC_50_ value of 2.1 nM. Moreover, it inhibited the proliferation of hematopoietic cancer cells, even with copy-amplified/mutated forms of CDK6 [134]. However, it is worth noting that their impact is still limited due to resistance development, which is the biggest challenge for PROTAC-based therapies at the moment [135].

As the effectiveness of traditional CDK8 inhibitors in the treatment of numerous cancers has yet to be confirmed, hence the need to elaborate new PROTACs for degrading the protein CDK8 became a driving force to overcome these shortcomings [136]. Cortistatin A was used to develop new derivatives. One of these compounds JH-XI-10-02 (**37**) is a potent CDK8 degrader (Figure 10). Its efficacy was verified by carrying out the degradation experiments in Jurkat and CRBN knockout Molt14 cells [137]. The synthesis of CDK8 degraders will definitely help to clarify whether targeting CDK8 is an effective strategy for treating cancer.

CDK9 forms a part of the positive transcription elongation factor b (P-TEFb) complex which together with cyclin T is responsible for the transcription elongation. CDK9 was found to be present in all tissues and numerous malignancies [138]. Due to the fact that CDK9 shares a high level of conservation sequence with other CDK members, it is difficult to obtain satisfactory selectivity [139]. In order to develop effective CDK9-targeting PROTAC it is necessary to identify lysine residues which can be targeted for ubiquitination and degradation [140]. The first selective CDK9 degrader **38** was developed on the basis of aminopyrazole derivative and Thalidomide (Figure 11). This CDK9 degrader reduced CDK9 protein activity in HCT116 cells by 56 and 65% at 10 and 20 µM, respectively, without affecting other CDKs [141]. Another CDK9 degrader, THAL-SNS-032 **39** was developed by conjugating pan-selective CDK inhibitor SNS-032 and Pomalidomide (Figure 11). It selectively degraded CDK9 with a 99% D_max_ at 250 nM in MOLT 4 cells after 6h treatment. Moreover, THAL-SNS-032 exhibited slower dissociation rates [142]. Yet another CDK9 degrader **40** was generated by conjugation of the natural compound Wogonin to Pomalidomide (Figure 11). This PROTAC induced the rapid degradation and showed more potency (IC_50_ = 17 ± 1.9 μM) than Wogonin (IC_50_ = 30 ± 3.5 μM) in MCF7 cells [143]. 

## 11. Conclusions

Cyclin-dependent kinases (CDKs) have unique tissue specific functions and dysregulation of CDKs and their cyclin partners is observed in a range of tumor types, and some of them have emerged as promising therapeutic targets in cancer. The major challenges in the CDK-targeted drug discovery are selectivity and bad responses, or resistance to treatments. However, the latest advances in the field provide encouragement that highly selective and potent inhibitors of human cyclin-dependent kinases with favorable pharmacokinetic properties will be identified.

## Figures and Tables

**Figure 1 ijms-22-02806-f001:**
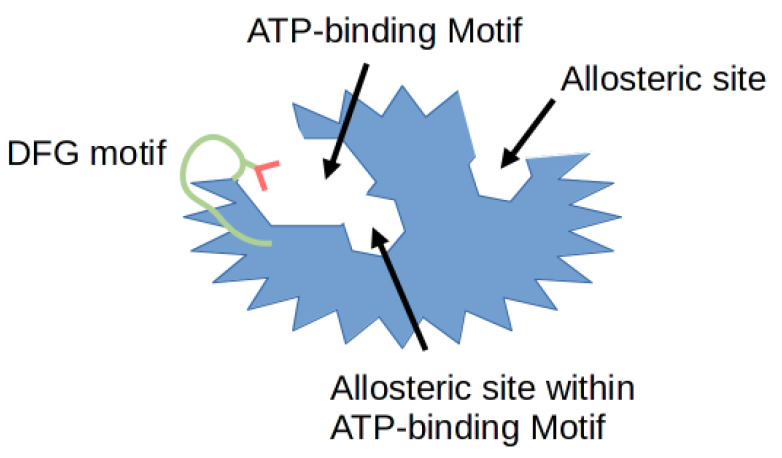
Schematic representation of different types of binding pockets. The protein kinase is shown in blue, with the Asp-Phe-Gly (DFG) motif in green. Red color denotes the aspartate amino acid residue of the DFG motif. The particular regions where different types of inhibitors bind are described below, the allosteric pocket is only a visualization and its place can be anywhere outside the ATP binding site.

**Figure 2 ijms-22-02806-f002:**
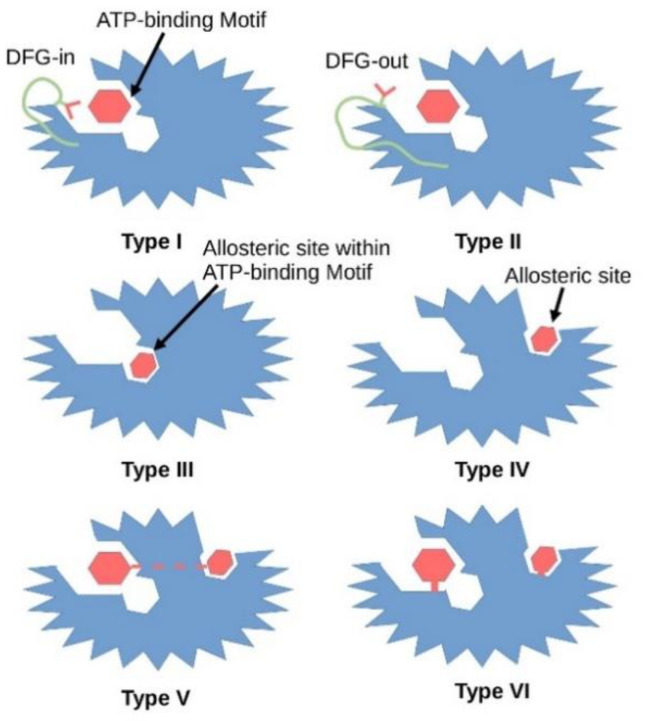
Graphical illustration of different types of kinase inhibitors and their mode of action. Dark red hexagon represents an inhibitor. The protein kinase is shown in blue, the DFG motif in green, the aspartate amino acid residue of the DFG motif in red. In 2015 Wu demonstrated that co-crystal structure of 3-phosphoinositide-dependent protein kinase 1 (PDPK1, PDK1) with ATP showed that type I inhibitors interact with the active conformation of the enzyme where the aspartate residue of the DFG motif points into the ATP binding pocket, while type II inhibitors stabilize the inactive conformation of the enzyme where the aspartate residue faces outward of the binding site (PDB entry: 4RRV). Type III inhibitors interact with the allosteric site within the ATP binding pocket. Type IV inhibitors interact with the allosteric site. However, the allosteric pocket is only a visualization and its place can be anywhere outside the ATP binding site. Type V inhibitors interact with both the allosteric and ATP binding pockets. Type VI inhibitors form covalent bonds with either the ATP binding pocket or the allosteric pocket.

**Figure 3 ijms-22-02806-f003:**
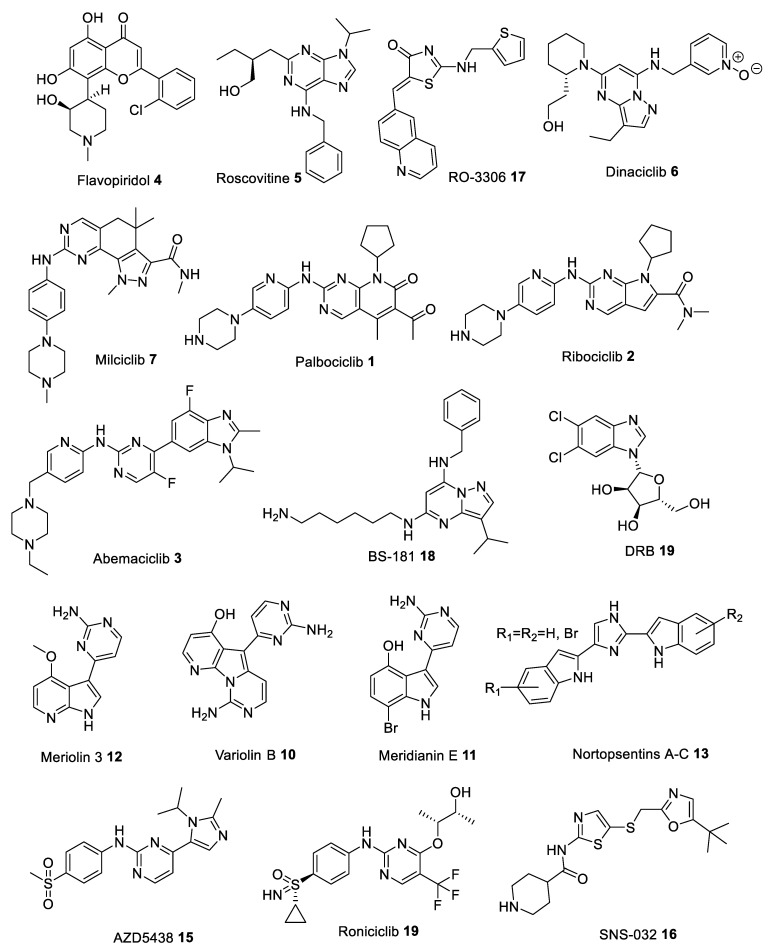
Chemical structures of some of the most studied type I cyclin-dependent kinase (CDK) inhibitors.

**Figure 4 ijms-22-02806-f004:**
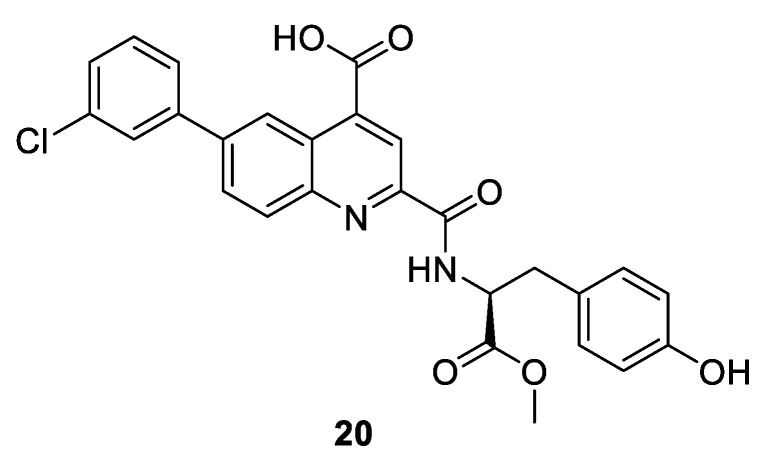
Chemical structure of quinoline-based type 1.5 inhibitor of monomeric CDK2.

**Figure 5 ijms-22-02806-f005:**
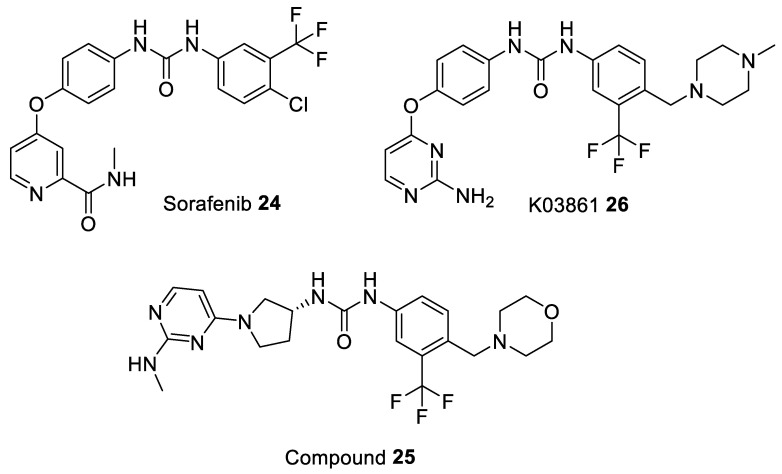
Chemical structures of some of the most studied type II CDK inhibitors.

**Figure 6 ijms-22-02806-f006:**
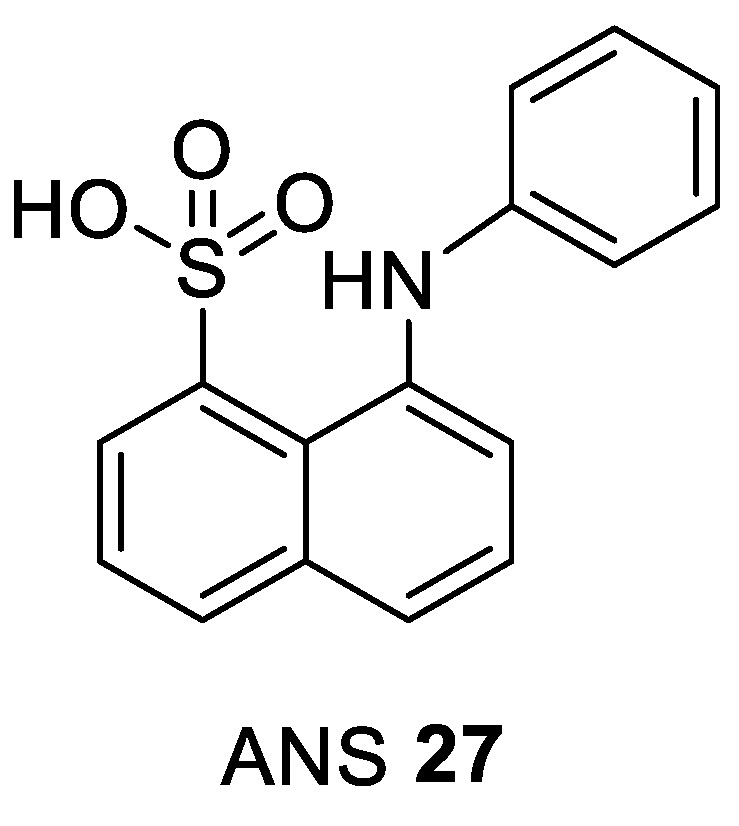
Chemical structure of 8-anilino-1-naphthalene sulfonate (ANS) the most studied type IV CDK inhibitor.

**Figure 7 ijms-22-02806-f007:**
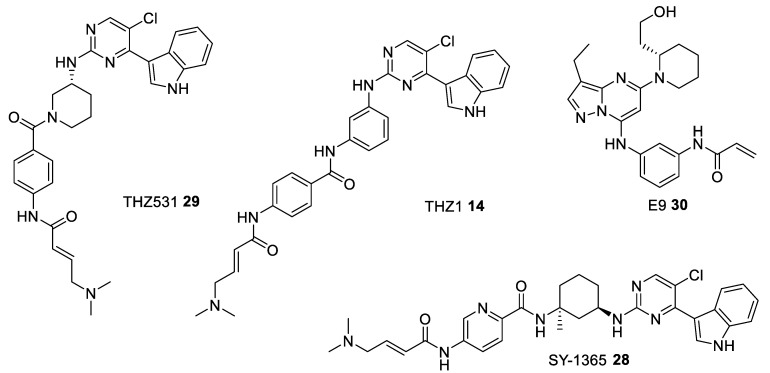
Chemical structures of some of the most studied type VI CDK inhibitors.

**Figure 8 ijms-22-02806-f008:**
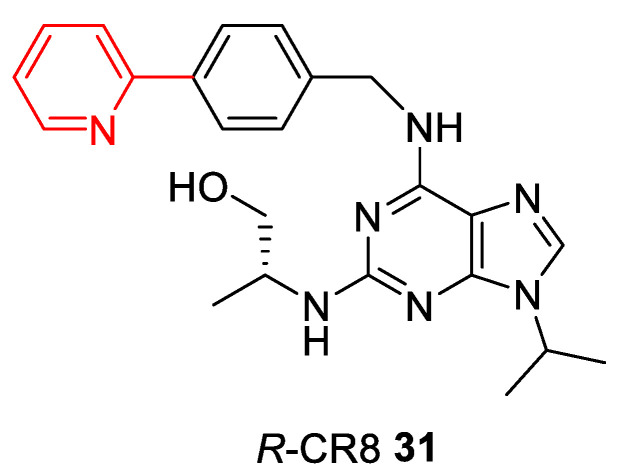
Chemical structure of CR8. A surface-exposed 2-pyridyl moiety of CR8 is responsible for glue degrader properties.

**Figure 9 ijms-22-02806-f009:**
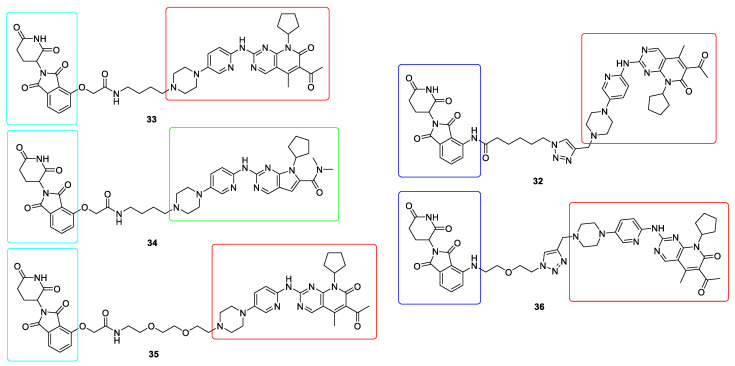
Chemical structures of CDK4/6 PROteolysis TArgeting Chimeras (PROTACs). Red rectangle denotes the palbociclib moiety, green rectangle denotes the ribociclib moiety, light blue rectangle denotes the thalidomide moiety and dark blue rectangle denotes the pomalidomide moiety.

**Figure 10 ijms-22-02806-f010:**
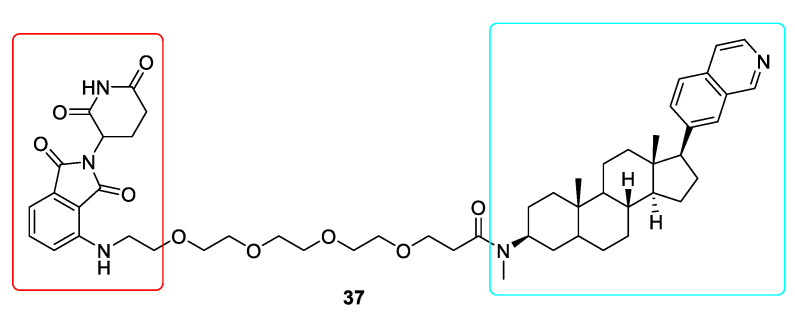
Chemical structure of CDK8 PROTAC. Red rectangle denotes the pomalidomide moiety, light blue rectangle denotes the Cortistatin A derivative JH-VIII-49 moiety.

**Figure 11 ijms-22-02806-f011:**
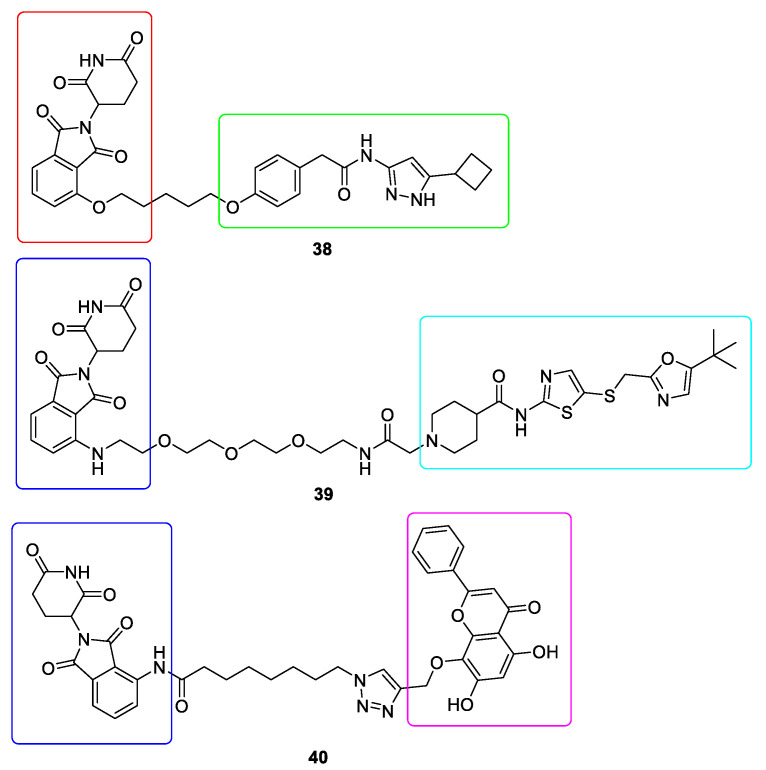
Chemical structures of CDK9 PROTACs. Red rectangle denotes the Thalidomide moiety, dark blue rectangle denotes the Pomalidomide moiety, green rectangle denotes the aminopyrazole derivative moiety, light blue rectangle denotes the SNS-032 moiety and violet rectangle denotes the Wogonin moiety.

**Table 3 ijms-22-02806-t003:** Selected biological data obtained from different assays with quinoline-based compound demonstrating that it targets monomeric CDK2.

Compound 2 20 [nM] [68]			
CDK2/A IC_50_	FP K_i_	TdCD K_d_	Clinical phase
>10,000	140	300	pre-clinical

**Table 4 ijms-22-02806-t004:** Type II CDK inhibitors under clinical evaluation.

	Kinase IC_50_ [nM]					Clinical Phase
	CDK2/A	Cyclin-Free CDK2	CDK4/D	CDK6/D	CDK8/C	
H-His-Ala-Lys-Arg-Arg-Leu-Ile-Phe-NH_2_ **21** [90]	140	-	-	-	-	pre-clinical
H-Ala-Ala-Abu-Arg-Arg-Leu-Ile-*p*FPhe-NH_2_ **22** [90]	80	-	-	-	-	pre-clinical
MM-D37K **23** [91,92]	-	-	active	active	-	phase I/II for Bladder, Gastrointestinal, Glioblastoma, and Malignant Melanoma
Sorafenib **24** [93]	-	-	-	-	74	drug approved for Renal Cell Carcinoma, Hepatocellular Carcinoma, AML, and Advanced Thyroid Carcinoma
Compound **25** [94]	-	-	-	-	17.4	pre-clinical
K03861 **26** [85]	10,000	9.7–50	-	-	-	pre-clinical

**Table 5 ijms-22-02806-t005:** Selected biological data obtained from different experiments with ANS which demonstrate that it targets monomeric CDK2.

ANS 27 [nM] [106]			Clinical Phase
CDK2/A IC_50_	Cyclin-Free CDK2 K_d_	ANS Displacement EC_50_	
91,000	37,000	600	pre-clinical

**Table 6 ijms-22-02806-t006:** Type VI CDK inhibitors under clinical evaluation.

	Kinase IC_50_ [nM]					Clinical Phase
	CDK2/A	CDK7/H/MAT1	CDK9/T1	CDK12/K	CDK13/K	
THZ531 **29** [118]	-	8500	10500	158	69	phase II—observational study for the patients-derived High Grade Serous Ovarian Cancer (HGSOC) organoids NCT04555473
THZ1 **14** [116]	-	3.2	-	>1000	>1000	pre-clinical
SY-1365 **28** [117]	-	20	-	-	-	phase I for Advanced Solid Tumors, Ovarian, and Breast Cancer NCT03134638
E9 **30** [119]	932	1210	23.9	-	-	pre-clinical

## Data Availability

Not applicable.

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
