# Peer review of "Inhibitors of Cyclin-Dependent Kinases: Types and Their Mechanism of Action"

_ijms, 2021, doi:10.3390/ijms22062806_

Round 1
Reviewer 1 Report
In this extensive review on CDK inhibitors, authors tried to cover all CDK inhibitors based on six different types of kinase inhibitors. The authors explained each inhibitor type with enough examples and consolidated information related to their pathways and partially touch upon their role in different disease states particularly cancer. However, some major changes would certainly help the readers for a better understanding of the content.
- The title and abstract are a bit misleading for the readers considering the content of the review. Advice would be to change both considering this review as a holistic overview on CDK inhibitors based on kinase inhibitor types and mechanism of action rather than pathways.
- A brief introduction on different CDKs and their role in the cell cycle or disease pathophysiology is missing (and maybe related graphics)
- Table 1 title should be changed as its mainly related to Type 1 CDK inhibitors potency against each CDK
- Table 1 and 2 format is different than other tables format and disrupt the consistency.
- Tables and figures are not well referred in the text
- Showing the chemical structure of the different type of CDK inhibitors in figures are not well explained as well as referred in the text
- The last line of the abstract regarding 'this review explains whether the ATP competitive inhibitors are the only option for future research': ThIs point was not clearly discussed within the review and not in the conclusions part of the review.
- Regarding the aforementioned point, the addition of recent technologies like transcriptional CDK inhibitors, such as target protein degradation (PROTACs), and their advances for the clinical evaluation would be an interesting and updated addition for the readers.
Author Response
Answers to the Reviewers
We would like to thank the Reviewers for precise and accurate reviews of our manuscript. Their recommendations helped us prepare a revised version of the manuscript. We included all remarks. We believe that the manuscript in the present form can be accepted by the Reviewers and Journal Editor.
Reviewer 1
In this extensive review on CDK inhibitors, authors tried to cover all CDK inhibitors based on six different types of kinase inhibitors. The authors explained each inhibitor type with enough examples and consolidated information related to their pathways and partially touch upon their role in different disease states particularly cancer. However, some major changes would certainly help the readers for a better understanding of the content.
- The title and abstract are a bit misleading for the readers considering the content of the review. Advice would be to change both considering this review as a holistic overview on CDK inhibitors based on kinase inhibitor types and mechanism of action rather than pathways.
Author’s answer: I have changed the title and abstract, please see the text.
- A brief introduction on different CDKs and their role in the cell cycle or disease pathophysiology is missing (and maybe related graphics)
Author’s answer: It has been added, please see the text.
- Table 1 title should be changed as its mainly related to Type 1 CDK inhibitors potency against each CDK
Author’s answer: It has been changed, please see the text.
- Table 1 and 2 format is different than other tables format and disrupt the consistency.
Author’s answer: The tables in the text body summarize the available information on the different types of CDK inhibitors. The format of the tables depends on the amount of information found about the described type of inhibitor.
- Tables and figures are not well referred in the text
Author’s answer: It has been changed, please see the text.
- Showing the chemical structure of the different type of CDK inhibitors in figures are not well explained as well as referred in the text
Author’s answer: It has been changed, please see the text.
- The last line of the abstract regarding 'this review explains whether the ATP competitive inhibitors are the only option for future research': ThIs point was not clearly discussed within the review and not in the conclusions part of the review.
Author’s answer: The abstract has been changed therefore there is no need to relate to this question.
- Regarding the aforementioned point, the addition of recent technologies like transcriptional CDK inhibitors, such as target protein degradation (PROTACs), and their advances for the clinical evaluation would be an interesting and updated addition for the readers.
Author’s answer: New chapter has been written, please see the text.
Reviewer 2 Report
The review by Lukasik et al. provides a comprehensive overview on CDK inhibitors and the various categories they fit in. It covers good depth of coverage of small molecules that are in clinic, in clinical trials or in pre-clinical development and guides readers to appropriate references for further details.
I only have few minor edits listed below to further improve this review prior to its consideration for a publication:
- Figure 1. The figure could be improved by adding a legend that annotates various elements (for e.g. hexagon denotes inhibitor) used by authors to explain different classes of inhibitors. Also the arrows used in the schematic need to be pointing accurately. For instance, for Type IV inhibitor the arrow is pointing to the inhibitor but the description says "Allosteric site".
- Line 62- 63. The statement needs a reference.
- Line 69-70. The reference review article does not cover the statement phrased by the authors in its entirety. It will be helpful if authors can bolster the statement by adding relevant citations.
- Line 70-71. Needs a reference.
- Line 76-78 Please rephrase the statement for clarity.
- Line 79-80 Need reference to corroborate authors statement.
- Line 84. The link listed for reference 17 is not functional. Please replace it with appropriate reference.
- Line 100. How is reference 23 linked to the statement?
- Line 103. One of the recommendation I have for authors is to reduce number of review articles as a citation for focused statements. It will be helpful if authors cite the actual study. For instance, please add actual study that showed DRB mediated inhibition of HIV transcription rather than using a review article.
- Line 108: Reference 28, Please cite the actual study than using the review article as a reference.
- Line 224-228. Add references.
- Line 359-360. Please rephrase for clarity.
Author Response
Answers to the Reviewers
We would like to thank the Reviewers for precise and accurate reviews of our manuscript. Their recommendations helped us prepare a revised version of the manuscript. We included all remarks. We believe that the manuscript in the present form can be accepted by the Reviewers and Journal Editor.
Reviewer 2
The review by Lukasik et al. provides a comprehensive overview on CDK inhibitors and the various categories they fit in. It covers good depth of coverage of small molecules that are in clinic, in clinical trials or in pre-clinical development and guides readers to appropriate references for further details.
I only have few minor edits listed below to further improve this review prior to its consideration for a publication:
- Figure 1. The figure could be improved by adding a legend that annotates various elements (for e.g. hexagon denotes inhibitor) used by authors to explain different classes of inhibitors. Also the arrows used in the schematic need to be pointing accurately. For instance, for Type IV inhibitor the arrow is pointing to the inhibitor but the description says "Allosteric site".
Author’s answer: It has been changed, please see the text.
- Line 62- 63. The statement needs a reference.
Author’s answer: It has been changed, please see the text.
- Line 69-70. The reference review article does not cover the statement phrased by the authors in its entirety. It will be helpful if authors can bolster the statement by adding relevant citations.
Author’s answer: It has been changed, please see the text.
- Line 70-71. Needs a reference.
Author’s answer: It has been changed, please see the text.
- Line 76-78 Please rephrase the statement for clarity.
Author’s answer: It has been changed, please see the text.
- Line 79-80 Need reference to corroborate authors statement.
Author’s answer: It has been changed, please see the text.
- Line 84. The link listed for reference 17 is not functional. Please replace it with appropriate reference.
Author’s answer: It has been changed, please see the text.
- Line 100. How is reference 23 linked to the statement?
Author’s answer: In the discussion part it is written about it
In vitro characterization of the kinases inhibited by DRB shows a 25-fold selectivity for CDK9 over both CDK7 and CDK2 ( 29, 46).
Another article: Cyclin-dependent kinase 9: a key transcriptional regulator and potential
drug target in oncology, virology and cardiology gives the same information. There is exactly the same statement.
At least in vitro DRB was reported to show 25-fold selectivity for CDK9 over both CDK7 and CDK2 [97].
- Line 103. One of the recommendation I have for authors is to reduce number of review articles as a citation for focused statements. It will be helpful if authors cite the actual study. For instance, please add actual study that showed DRB mediated inhibition of HIV transcription rather than using a review article.
Author’s answer: It has been changed, please see the text.
- Line 108: Reference 28, Please cite the actual study than using the review article as a reference.
Author’s answer: It has been changed, please see the text
- Line 224-228. Add references.
Author’s answer: It has been changed, please see the text
- Line 359-360. Please rephrase for clarity.
Author’s answer: It has been changed, please see the text
Round 2
Reviewer 1 Report
In the revised manuscript the authors adequately addressed all the issues and concerns raised in the previous version, particularly for broadening the readership.
Amendments to the title and abstract are currently justifying the review content. Besides, the addition of sections on CDKs introduction and Targeted Protein Degradation (TPD) certainly make this review more interesting and updated.
Overall, the authors addressed all my concerns.